## [Transparent Peer Review file · Nature Communications]

The Lolal-dpp axis mediates the regulation of host reproduction by gut symbionts in insects

Corresponding Author: Professor Hongyu Zhang

Version 0:

Reviewer comments:

Reviewer #1

(Remarks to the Author)

The authors investigated the impact of gut symbionts on the reproductive capacity of the fruit fly *Bactrocera dorsalis*. Its gut harbors a stable microbial community dominated by Enterobacteriaceae, among which *Enterobacter hormaechei* plays a key role. Through a combination of approaches—including gut microbiota manipulation, metabolomic and proteomic profiling, RNA interference, quantitative RT-PCR, and microscopy—they demonstrated that *E. hormaechei* promotes ovarian development and egg production. Mechanistically, the bacterium produces nicotinic acid (NA), which enhances NAD synthesis and ATP production, thereby activating the ubiquitin–proteasome system (UPS). The UPS promotes ubiquitination and degradation of the transcription factor Lolal, preventing its excessive accumulation. Proper Lolal balance is essential: excess Lolal in bacteria-depleted flies triggers decapentaplegic (dpp) overexpression and suppresses ovarian development, while insufficient Lolal reduces dpp expression, also impairing reproduction. The study reveals that gut bacteria-derived NA maintains host protein homeostasis and ensures female reproductive capacity.

I believe the study is of general interest to the insect symbiosis community. However, I have several suggestions that could improve the manuscript. In particular, I have major comments regarding the statistical analyses, the result description, and their interpretation.

Major comments

1- I have some concerns about the statistical analyses. In my opinion, the tests used may not be the most appropriate, given the data. Is your data normally distributed for use with the t-test and ANOVA test? I am not sure that ANOVA is the best test for percentage or relative content data. I recommend considering the use of general linear models and/or generalized linear models, depending on the data distribution. After fitting the models, you can conduct pairwise comparisons for the significant variable.

In the materials and methods section, please review your analyses and explain the methods used and the variables tested for each experiment. Moreover, you need to create tables in the supplementary data that include all relevant statistical results (e.g., test values, degrees of freedom, p-values for models and pairwise comparisons).

In the results section, you need to reference your statistical results. Every time you describe a result, please include the test name, test value, degrees of freedom, p-values, and refer to supplementary tables.

I have another concern about the ovarian development grade results. You explain throughout the study that there are differences in ovarian development between different conditions (Fig. 1e and l, Fig. 3g and j, Fig. 5f). However, none of these are supported by statistical tests. Please test this statistically.

2- The results need to be further described. It is necessary to compare the different treatments with controls. Moreover, the results are also sometimes overinterpreted. For example, in line 113, you indicate that the dietary NA supplementation

rescued ovarian development. But this is not the case. This partly rescued ovarian development (verify with statistical analysis).

I suggest verifying these throughout the result section and subsequently adjusting them. I also recommend combining the results and discussion into a single section, with a general conclusion.

3- The authors demonstrated the mechanism involved in the fly ovarian development and the number of eggs produced by females. However, I do not entirely agree with the title and conclusion regarding the impact on the host reproduction. To do so, the authors need to study the number of viable individuals produced, i.e., egg hatching rate, total number of eggs, and larval survival rate. They should therefore add these new parameters or further qualify their conclusions.

4- Have you tested the fly survivorship after bacterial removal and recolonization? Your study must determine whether the absence of symbionts only impacts the development of ovaries and eggs or whether it also affects the fly survivorship. It is also necessary to know whether recolonization is effective and has the same effect on flies as the control.

Comments by section

Abstract

Please add information about the model used in the study.

Introduction

You begin your introduction by focusing on gut symbionts, but then you include examples involving insect endosymbionts. You need to make a better distinction between vertically transmitted endosymbionts and gut microbes, and not give examples of endosymbionts if you are talking about gut microbiota. It would therefore be relevant to compare their roles in insects, particularly in their reproduction.

I recommend adding more information on the regulatory mechanisms known to be involved in insect reproduction (intracellular pathways and hormonal signals) and the impact of gut microbes. In particular, I would like to see more about other closely related models such as *Drosophila melanogaster* and *Bactrocera oleae*.

I suggest improving the transition between the third and fourth paragraphs. The molecular mechanisms by which gut symbionts regulate insect reproduction are not well understood, and ubiquitination could be one of them.

Materials and Methods

L 399: Please provide more information about *Bactrocera dorsalis*, particularly regarding the microbial composition of its gut. Additionally, I wonder whether the flies carry *Wolbachia*? If so, where are they located, and what is their impact on the fly? Have you checked whether eliminating gut bacteria also eliminates this bacterium? Do you think the bacterium could affect your study?

L 407: Regarding bacterial elimination and recolonization, could you provide more information on which bacteria were reintroduced and how they were reintroduced? Did you introduce all the bacteria from the gut flies directly, or did you cultivate the bacteria and reintroduce them in the same quantity? And when you introduced only the *E. hormaechei* strain, how much did you introduce (CFU)?

L 424: Regarding the assessment of gut bacterial load, please specify which 16S rRNA primer pairs you used. Are they universal primers or specific to each bacterium present in this insect?

L 451: Further clarification is needed regarding the study of ovarian development stages. When did you take the measurements (female age)? And were they identical for each experiment?

Please add descriptive titles for the tables in the supplementary data. Furthermore, additional information on primer pairs needs to be added to Table S2 in the supplementary materials.

Results

L 111: Have you tried adding all the metabolites together? Mixing all these metabolites could complete the ovarian development.

L 114: Please explain why you tested for vitellogenin deposition.

LL 132-136: Please split this sentence into two and add that NA is undetectable in hemolymph and ovaries.

L 167: Please explain why you checked mitochondrial oxidative phosphorylation.

LL 168-169: I disagree. The results show the opposite. Perhaps this is a color issue?

L 200: Please explain why you silenced the expression of *Uba 1*.

L 276-278: Can you explain why the control in Supplementary Figure 11 is similar to ABX+dsGFP? Furthermore, I believe that Figure 11 should be included in Figure 5.

L 282: Please explain why there is no effect on the length of mature eggs.

L 290: Why are Figure 5a and Figure S12a different?

Figures

Figure 1: I find the figures too small and difficult to read. I would recommend changing it to portrait format and enlarging all the figures it contains.

Figure 2: Figures 2d and j must be adjacent because they are identical.

Figure 4: Figures should be positioned in order.

Remarks on code availability

Raw data and scripts are not available.

Proteomic and metabolomic data have to be deposited in a public data repository.

Reviewer #2

(Remarks to the Author)

This study uses *Bactrocera dorsalis* as a model system to investigate how gut symbiotic microbes influence host insect reproduction (ovarian development), not only from the perspective of nutritional metabolism but also in terms of gene expression regulation. The authors first performed RNA-seq analyses and discovered that niacin (vitamin B3), a metabolite produced by gut bacteria, may play a crucial role. The authors then demonstrated that microbial NA indeed affects ovarian development by re-infecting antibiotic-treated flies with gut bacteria, as well as by infecting them with bacterial mutants deficient in NA biosynthetic genes.

Beyond such experimental validation, the authors further examined the mechanism by which vitamin B3 regulates ovarian development. Using detailed biochemical and reverse genetic approaches (RNAi), they showed that reduced NA levels lead to decreased mitochondrial activity, which in turn reduces ubiquitination of Lolal, a regulatory factor of the growth factor decapentaplegic (dpp). As a result, dpp expression increases, ultimately suppressing ovarian development in the insect. Although symbiotic bacteria are known to contribute significantly to ovarian development and reproduction in insects, the underlying mechanisms (or causal relationships) have remained unclear. This study partly elucidates how bacterial metabolites influence insect gene regulation and lead to reduced ovarian development. Although, the manuscript is generally well written and well organized, I still have several major and minor concerns, as outlined below.

Major Concerns

1. Gut microbiota characterization

No information is provided regarding the gut microbiota of conventionally reared *Bactrocera dorsalis*. Previous studies have shown that the composition and diversity of gut microbiota can be influenced by both host strain and food resources (Tian et al. 2023; *Frontiers in Microbiology*). Including background data on the gut microbiota—such as NGS results—is critical for interpreting and validating the downstream experiments.

2. Selection of bacterial strains

How diverse are the microbes that colonize the midgut of *B. dorsalis*? And how were the five bacterial strains used in this study selected? Were only these five strains recovered through culturing (L121–126)? Providing a rationale based on the actual composition of the gut microbiota is necessary to justify the experimental design.

3. Assessment of reproductive ability

To evaluate female reproduction, the authors measured ovarian development, mean egg length, and total egg number. Did gut microbiota influence only ovarian development without affecting the timing of oviposition? Enhanced ovarian development could plausibly accelerate oviposition. Clarification would strengthen the conclusions.

4. Interpretation of dsRNA experiments (Fig. 5)

The authors tested that bacteria-derived NA modulates the Lolal–dpp signaling pathway to influence ovary development. However, in the RNAi experiments, it is still difficult to conclude that silencing Lolal or dpp rescues ovary development. In fact, ovarian size and maturation do not appear to be restored (Fig. 5e, f), and overall fecundity is significantly reduced. Although the authors provide some explanations (L290–303), these data do not convincingly support the claim that Fig. 5 provides mechanistic evidence for microbial regulation of ovary development.

5. Statistical analysis

Statistical details are insufficient. Throughout the manuscript, values such as t-values, F-values, and degrees of freedom are missing. For two-way ANOVA analyses, the interaction term between factors is important for assessing treatment effects on fecundity, yet this is not described. In addition, there is no explanation of how multiple comparisons were performed. In figures such as Fig. 1g, significance is indicated with letters, but the statistical test and significance level are not stated. The reporting of statistical analyses and figure legends is inadequate. If statistical methods beyond ANOVA were used, they must be described explicitly.

Minor Comments

- L19 and L26: These two sentences are identical. In the abstract, this is redundant.
- L44: “B” vitamins? Please clarify.
- L52: The phrase “genetic advantage” is unclear.
- L100: Consider rephrasing as “a key metabolite.”

- L102: The authors describe “metabolomic analysis,” but the experiment appears to be transcriptomic (RNA-seq).
- L135: The authors state bacterial growth is unaffected. Does this apply only in nutrient-rich media? Wouldn't growth be impaired in minimal medium?
- L184: The claim that these results indicate reduced UPS activity seems logically weak. A supporting reference is needed.
- L232: There is a logical gap. What about the other 26 genes—are they irrelevant?
- L262: The subject would read more naturally as “gut bacteria” rather than “Lolal.”
- L294: Is 1.09 mm the mean value? If so, the standard deviation should also be included.
- L355: Should read “Vitamin B3 deficiency.”
- L372: Define “GSC.”
- L436: Yeast extract typically contains vitamin B3. If it was used as a food source, why does deficiency occur?
- L459: Please clarify “120 minutes.”
- L543: Information on primer sets used for dsRNA synthesis should be provided.
- L565: Please specify the plasmid into which Lolal was cloned.
- L580: The antibody production method is unclear. If a commercial source was used, please state the supplier.
- Fig. 1g, i, j: The y-axis label (“NA related content”) should include units.

Reviewer #3

(Remarks to the Author)

Qiao and colleagues have investigated the relationship between the gut microbiota, ovarian development and consequent fertility in the oriental fruit fly, *Bactrocera dorsalis*. They showed that gut symbionts, particularly *Enterobacter hormaechei*, produce nicotinic acid which improves mitochondrial function and promotes the ATP-mediated ubiquitination and consequent degradation by the proteasome pathway of a transcription factor, Lolal, regulating the expression of *dpp* in ovaries.

This is a well written manuscript and a thorough investigation of molecular interactions between gut symbionts and ovarian development, using an impressive and complementary methods palette to assemble a coherent mechanistic picture. I don't have technical concerns. The experiments seem to have been well-performed, are consequential and complementary, and together support the mechanistic claims. I am excited about this study and have only a few general recommendations to add an evolutionary angle to the interpretation of these results.

The manuscript would benefit from discussing why ovarian development and fertility should depend so strongly on gut microbial signals. What is the evolutionary rationale for such dependence, given that imperfect transmission or acquisition of gut symbionts could severely reduce reproduction? Could the authors add some introduction and discussion on the evolutionary implications of their findings and explain whether such connection makes sense from an evolutionary perspective? The study focuses on ovarian development and didn't touch upon other potential developmental effects of these NA-Lolal-*dpp* interactions. Are these effects specific to reproduction or by-products of nutritional deficiencies hampering development more in general? Given the loss of fertility, these insects would not do well if a specific set of symbionts is not faithfully acquired or is subsequently disrupted. While the introduction already states that *Bactrocera dorsalis* flies host a stable and complex microbial community dominated by Enterobacteriaceae, a bit more background of this would be helpful. Are *Enterobacter hormaechei*, or other symbionts capable of producing NA, always present within the gut of individual flies in nature? How are they acquired? Are they vertically transmitted? Can they be lost?

A few minor suggestions:

Some statements in the abstract and introduction are too broad and should be made more conditional. Line 16: not all commensal microbiota shape reproduction and not in every animal, please modify to “..microbiota can play an integral role...”. Similarly, in Line 35: “The insect gut can comprise complex and diverse symbiotic systems”. Line 41: “..this gap can be filled by gut microbiota”. Line 55: “..microbiota can be associated with insect reproductive...”

Lines 225 and 226: This may be due to my limited knowledge of this specific technique, but I did not fully understand how the 819 and 665 proteins were counted and relate to the numbers given in the scatterplots in Figures 4b and 4c. There is also no explanation in the figure caption about what these numbers (e.g., “1(652)” at the top left of panel b) mean. Could you please improve clarity in this passage?

It is also not completely clear to me how the authors moved from studying proteome-wide ubiquitination and protein abundance to focusing specifically on Lolal. I understand that the analysis showed that 27 proteins are both hypo-ubiquitinated and upregulated in the comparison between ABX and ctrl and hyper-ubiquitinated and downregulated in the comparison between EH and ABX. The next step then shows that Lolal is among about 20 proteins presented in Supplementary Figure 8b, but I could not understand how the role of these proteins, and only these, in development was determined. A more precise clarification of how this “development” set was identified, its overlap with the 27 proteins above, and the reasoning for focusing on Lolal specifically would help here, as this is a key step.

Version 1:

Reviewer comments:

Reviewer #1

(Remarks to the Author)

The authors have addressed my concerns and made the necessary changes. Therefore, I consider the modified paper suitable for publication.

However, I have one observation regarding the statistical significance in the tables of the Source Data file. The authors should be attentive when interpreting the pairwise comparisons. I identified three issues, but I recommend a thorough check before publication:

1. Figure 1J: The comparison between Ctrl and ABX+EH shows no significant difference.
2. Figure 1K (ovary): Only ABX and ABX+EH show a significant difference.
3. Figure 5L: There is a significant difference between ABX+dsLolal and ABX+dsdpp.

Additionally, in Figure 5I, the statistical analysis is missing.

Reviewer #2

(Remarks to the Author)

I appreciate the authors' thorough and sincere response, including the additional data. The new analyses convincingly demonstrate that the gut microbiome is indeed extremely simple and stable. I also find the revised description of the "partial rescue" appropriate and mentioned. In addition, the recombinant protein injection experiments are straightforward and very well executed (Although I still have a minor concern regarding potential immune responses, this does not affect my overall assessment). Overall, these additional experiments and clarifications have fully addressed my previous concerns.

Reviewers' comments:

Reviewer #1 (Remarks to the Author): The authors investigated the impact of gut symbionts on the reproductive capacity of the fruit fly *Bactrocera dorsalis*. Its gut harbors a stable microbial community dominated by Enterobacteriaceae, among which *Enterobacter hormaechei* plays a key role. Through a combination of approaches—including gut microbiota manipulation, metabolomic and proteomic profiling, RNA interference, quantitative RT-PCR, and microscopy—they demonstrated that *E. hormaechei* promotes ovarian development and egg production. Mechanistically, the bacterium produces nicotinic acid (NA), which enhances NAD synthesis and ATP production, thereby activating the ubiquitin–proteasome system (UPS). The UPS promotes ubiquitination and degradation of the transcription factor Lolal, preventing its excessive accumulation. Proper Lolal balance is essential: excess Lolal in bacteria-depleted flies triggers decapentaplegic (*dpp*) overexpression and suppresses ovarian development, while insufficient Lolal reduces *dpp* expression, also impairing reproduction. The study reveals that gut bacteria-derived NA maintains host protein homeostasis and ensures female reproductive capacity.

I believe the study is of general interest to the insect symbiosis community. However, I have several suggestions that could improve the manuscript. In particular, I have major comments regarding the statistical analyses, the result description, and their interpretation.

Answer: We sincerely thank the reviewer for these helpful comments. By following suggestions, we have revised the manuscript and supplemented it with additional analyses and new reproductive parameters, thereby substantially strengthening the study.

Major comments

1- I have some concerns about the statistical analyses. In my opinion, the tests used may not be the most appropriate, given the data. Is your data normally distributed for use with the *t*-test and ANOVA test? I am not sure that ANOVA is the best test for percentage or relative content data. I recommend considering the use of general linear models and/or generalized linear models, depending on the data distribution. After fitting the models, you can conduct pairwise comparisons for the significant variable.

Answer: We sincerely appreciate the reviewer's valuable comments and suggestions. We first applied the Shapiro–Wilk test to assess data normality and the Brown–Forsythe test/*F*-test to evaluate homogeneity of variance. If the data met the assumptions of normality and equal variance, parametric tests, such as unpaired *t*-test and one-way ANOVA followed by Tukey's multiple comparisons test, were used to compare two groups and multiple groups, respectively. If not, nonparametric tests, such as Mann–Whitney *U*-test and Kruskal–Wallis test followed by Dunn's multiple comparisons test, were used.

For relative content data, egg hatching rate, these are considered continuous variables. Most of them in our study are approximately normally distributed, not near the boundaries (0% or 100%), exhibited homogeneity of variance. So, *t*-test or ANOVA is suitable for them. For some of them that not normally distributed, non-parametric alternatives (Mann–Whitney *U*-test or Kruskal–Wallis test) were used. For percentage data describing ovarian developmental grades, which represent count frequency of categorical variables, the Pearson Chi-square test followed by pairwise comparisons is appropriate, as categorical variables are discrete and cannot be described by means or variances.

We believe these statistical approaches in new manuscript are appropriate for our data structure and adequately address differences among groups. All data and detailed descriptions of the analytical procedures are available in the Source Data file submitted to the editor. Thank you again for these valuable comments.

In the materials and methods section, please review your analyses and explain the methods used and the variables tested for each experiment. Moreover, you need to create tables in the supplementary data that include all relevant statistical results (e.g., test values, degrees of freedom, p-values for models and pairwise comparisons).

Answer: Thank you for your suggestion. The *Statistical Analysis* section has been revised to clearly describe the variables analyzed and the statistical tests applied in each experiment (Please see Lines 797-812 in new manuscript).

In addition, we have created a table in the *Source Data* file to summarize the statistical analyses used for each dataset, including: results of data normality and homogeneity of variance, the test methods, test values, degrees of freedom, p-values, and adjusted p-values for each pairwise comparison, as an example shown in Table R1 (below). The *Source Data* file has been submitted to the editor.

Hemolymph					
	Ctrl	ABX	ABX+gm		
	0.978947	0.179211	0.455263		
	0.784211	0.418421	0.778947		
	0.694737	0.265789	0.942105		
	1	0.203947	0.510526		
Test for normal distribution		Ctrl	ABX	ABX+gm	
Shapiro-Wilk test					
W		0.8791	0.8816	0.9135	
P value		0.3347	0.3457	0.501	
Passed normality test (alpha=0.05)?	Yes	Yes	Yes		
P value summary	ns	ns	ns		
Brown-Forsythe test					
F (DFn, DFd)	2.977 (2, 9)				
P value	0.1018				
P value summary	ns				
Are SDs significantly different (P < 0.05)?	No				
ANOVA summary					
F	12.94				
P value	0.0022				
P value summary	**				
Significant diff. among means (P < 0.05)?	Yes				
R square	0.742				
ANOVA table	SS	DF	MS	F (DFn, DFd)	P value
Treatment (between columns)	0.7443	2	0.3722	F (2, 9) = 12.94	P=0.0022
Residual (within columns)	0.2588	9	0.02875		
Total	1.003	11			
Tukey's multiple comparisons test	Mean Diff.	95.00% CI of diff.	Significant?	Summary	Adjusted P Value
Ctrl vs. ABX	0.5976	0.2629 to 0.9324	Yes	**	0.002
Ctrl vs. ABX+gm	0.1928	-0.1420 to 0.5275	No	ns	0.2915
ABX vs. ABX+gm	-0.4049	-0.7396 to -0.0701	Yes	*	0.0202

Table R1. An example for showing relevant statistical results in *Source Data* file

In the results section, you need to reference your statistical results. Every time you describe a result, please include the test name, test value, degrees of freedom, p-values, and refer to supplementary tables.

Answer: Thank you for this insightful comment. We have incorporated these statistics into all the result descriptions. For example, Line 98: “gut bacteria decreased by 95% (unpaired t -test, $t(4) = 79.0$, $p < 0.0001$)” and Lines 124-126: “ABX females exhibited a 75% decrease in fecundity compared with Ctrl females, whereas NA supplementation significantly increased egg production to 3.0-fold of ABX females (one-way ANOVA, $F(2, 21) = 43.6$, $p < 0.0001$; Fig. 1f)”. Test values for t -test and one-way ANOVA are presented as $t(df)$ and $F(DFn, DFd)$ (DFn: numerator degrees of freedom; DFd: denominator degrees of freedom). In addition, we have indicated in each figure legend that “Source data are provided as a Source Data file”.

I have another concern about the ovarian development grade results. You explain throughout the study that there are differences in ovarian development between different conditions (Fig. 1e and l, Fig. 3g and j, Fig. 5f). However, none of these are supported by statistical tests. Please test this statistically.

Answer: We sincerely appreciate your valuable suggestion. We totally agree that differences in ovarian development should be supported statistically. Statistical analyses have been performed, and they support our original conclusions.

To do so, we performed Pearson Chi-square test to analyze differences in ovarian development grades among different treatments. When the overall test was significant, post-hoc pairwise comparisons (Chi-square partitioning) were performed with adjusted p -values calculated using the Bonferroni correction. The Chi-square values (χ^2), degrees of freedom, and p -value for each experiment have been updated in the corresponding *Results* section and *Source Data* file. In the corresponding figure, we have added letters to indicate pairwise differences, and the degree of these differences is described in results and figure legend. This method is also described in the *Materials and Methods* section: “Differences in ovarian development grades were evaluated using Pearson’s Chi-square test followed by post-hoc pairwise comparisons (Chi-square partitioning) with Bonferroni correction” (New Line 810).

2- The results need to be further described. It is necessary to compare the different treatments with controls. Moreover, the results are also sometimes overinterpreted. For example, in line 113, you indicate that the dietary NA supplementation rescued ovarian development. But this is not the case. This partly rescued ovarian development (verify with statistical analysis). I suggest verifying these throughout the result section and subsequently adjusting them. I also recommend combining the results and discussion into a single section, with a general conclusion.

Answer: We sincerely appreciate the reviewer’s suggestion. We have added more detailed descriptions of results throughout the manuscript, particularly regarding the magnitude of changes rather than simply indicating increases or decreases. Pairwise comparisons with appropriate groups have also been explicitly described, and statistical significance is now indicated in the text (please refer to the revised manuscript).

For the overinterpreted results, we have revised it and the sentence now reads: “dietary NA supplementation partly rescued ovarian development” (New Line 121). We have checked the manuscript throughout to prevent any overinterpretation. This includes: “dietary NA only promoted development in 9% of the ovaries to grades IV–V” (Line 244), and “This resulted in a partial rescue of ovarian development” (Line 327). Thank you again for careful observation.

Regarding the structure of results and discussion, according to the submission guideline of *Nature Communication*, the Results and Discussion should be presented as separate sections. We fully understand the reviewer's concern regarding the separation of these sections; however, in our manuscript, the Discussion was structured to closely follow and interpret the main findings presented in the Results. For example, in Discussion Lines 393-421, we discussed the interdependent symbiotic relationship between microbiota and their host insects during co-evolution, with particular emphasis on their influence of reproduction; Our study revealed that the gut microbiota influence the host ubiquitination level. As in the Discussion section (Lines 377-392), we focused on the topic of microbial regulation of host insect's gene expression at multiple levels, including histone methylation, acetylation, and DNA methylation, which exert broad regulatory effects on host gene networks; In addition, we specifically discussed the roles of *Lolal* and *dpp* in reproductive development in Lines 438-464. Therefore, it may be more appropriate to present the Results and Discussion as two separate sections.

3- The authors demonstrated the mechanism involved in the fly ovarian development and the number of eggs produced by females. However, I do not entirely agree with the title and conclusion regarding the impact on the host reproduction. To do so, the authors need to study the number of viable individuals produced, i.e., egg hatching rate, total number of eggs, and larval survival rate. They should therefore add these new parameters or further qualify their conclusions.

Answer: Thank you for your suggestions. We fully agree that the conclusion regarding the impact on the host reproduction cannot be fully supported by only ovarian development and egg production. As suggested, we have now added data on egg hatching rate and larval survival rate throughout the study to strengthen our conclusions. Our results show that the egg hatching rate of ABX females was reduced compared with the control, but was significantly improved after NA supplementation and *E. hormaechei* recolonization (Figure R1a-e, below). In addition, compared with dsGFP-injected females, *Lolal* and *dpp* RNAi reduced egg hatching rates (Figure. R1f, below). However, the depletion of gut bacteria, NA supplementation, or *Lolal-dpp* pathway interference did not affect larval-to-adulthood survival rate (Figure R1g, h below).

Bactrocera dorsalis exhibits a relatively long lifespan, surviving over 100 days under laboratory conditions, and the female continuously lays eggs throughout its lifespan. Consistent with previous studies^{1,2}, we have now reported accumulated egg production within our experimental time window (Figure R2b-g, below). The reduced egg production observed in gut bacteria-depleted females results from developmentally suppressed ovaries. Moreover, our findings suggested that *Lolal-dpp* axis is critical for the microbiota-mediated regulation of reproduction. As *dpp* has been demonstrated to be an important initial factor in regulating insect reproduction³, we believe our results provide strong support for the role of gut microbiota in host reproduction though we did not quantify lifetime egg production. We have clarified this in the Discussion: "The observed reduction in fecundity resulted from suppressed ovarian development, which was caused by insufficient NA-induced impairment of ATP production and UPS activity. This phenotype is expected to persist if the gut microbiota or dietary NA remains absent over an extended period" (Lines 431-434).

The new data have been incorporated into the revised manuscript (Please see the New Figures: Fig. 1f, g, n and o, Fig. 3h, i, l and m, Fig. 5m and n, Supplementary Fig. 3g, Supplementary Fig. 13f-h). Thanks again for these constructive suggestions.

Figure. R1 Egg hatching rate. (a, b) NA supplementation (a) and *E. hormaechei* recolonization (b) promoted egg hatching rate. $n = 8$ for each. (c) *Uba1* RNAi decreased egg hatching rate. $n = 8$ for each. (d) *Uba1* RNAi in NA-supplemented ABX females couldn't rescue egg hatching rate. $n = 8$ for each. (e) *Lolal* RNAi and *dpp* RNAi in ABX females partly restore egg hatching rate. $n = 8$ for each. (f) *Lolal* RNAi and *dpp* RNAi in normal females decreased egg hatching rate. $n = 8$ for each. (g, h) Antibiotic treatment and NA supplementary (g) and *Lolal* RNAi and *dpp* RNAi (h) have no effect on larval survival rate. $n = 8$ for each. (a, b, d, e, f, g, h) one-way ANOVA followed by Tukey's multiple comparisons test, for (a, b, d, e), $p < 0.0001$; for (f), $p < 0.001$; for (g, h), $p > 0.05$; (c) Mann-Whitney test. $**p < 0.01$; ns, non-significant.

Figure. R2 Number of eggs laid. (a, b) NA supplementation (a) and *E. hormaechei* recolonization (b) promoted egg production in ABX females. $n = 8$ for each. (c) *Uba1* RNAi decreased egg production. $n = 8$ for each. (d) *Uba1* RNAi in NA-supplemented ABX females couldn't rescue egg production. $n = 8$ for each. (e) *Lolal* RNAi and *dpp* RNAi in ABX females partly restore egg production. $n = 8$ for each. (f) *Lolal* RNAi and *dpp* RNAi in normal females decreased egg production. $n = 8$ for each. (a, b, d, e, f) one-way ANOVA followed by Tukey's multiple comparisons test, $p < 0.0001$; (c) unpaired *t*-test; **** $p < 0.0001$.

4- Have you tested the fly survivorship after bacterial removal and recolonization? Your study must determine whether the absence of symbionts only impacts the development of ovaries and eggs or whether it also affects the fly survivorship. It is also necessary to know whether recolonization is effective and has the same effect on flies as the control.

Answer: Thanks for your suggestion. To address this concern, we examined adult survivorship following gut bacterial removal and recolonization. Our results showed that antibiotic treatment did not significantly affect adult survival compared with the control group. Recolonization with the total culturable gut bacteria effectively restored the gut bacterial load and this bacteria reinfection also had no effect on fly survivorship (Figure R3, below).

The new data have been incorporated into the revised manuscript (Please see Lines 100-103 and the New Figures: Supplementary Fig. 2b, c).

Figure. R3 Bacterial removal and recolonization did not affect female’s survivorship. (a) Gut bacterial load after antibiotic treatment and total culturable gut bacterial recolonization. $n = 4$; one-way ANOVA followed by Tukey’s multiple comparisons test, $p < 0.0001$. (b) Survival curves by log-rank (Mantel–Cox) analysis (individual = 80); ns, non-significant.

Comments by section

Abstract

Please add information about the model used in the study.

Answer: Thanks for your suggestion. We have revised the Abstract accordingly. The revised sentence now reads: “Here, we report that gut bacteria promote host reproduction in the oriental fruit fly *Bactrocera dorsalis* (Diptera: Tephritidae), an important agricultural pest” (Lines 16-17).

Introduction

You begin your introduction by focusing on gut symbionts, but then you include examples involving insect endosymbionts. You need to make a better distinction between vertically transmitted endosymbionts and gut microbes, and not give examples of endosymbionts if you are talking about gut microbiota. It would therefore be relevant to compare their roles in insects, particularly in their reproduction.

Answer: We thank the reviewer for pointing out this mistake. We have removed the examples of endosymbionts from the Introduction, particularly in the second and third paragraphs, and added examples relevant to gut microbiota instead. In addition, we carefully checked other parts of the manuscript to ensure a clear distinction between endosymbionts and gut microbes.

We have added a comparison between gut microbiota and endosymbionts, regarding their roles in influencing insect reproduction (Please see Lines 409-421 in the revised manuscript).

I recommend adding more information on the regulatory mechanisms known to be involved in insect reproduction (intracellular pathways and hormonal signals) and the impact of gut microbes. In particular, I would like to see more about other closely related models such as *Drosophila melanogaster* and *Bactrocera oleae*.

Answer: Thank you for the suggestion. We agree that this addition is important and have revised the Introduction as follows (Lines 51-61):

“The reproductive process in female insects is crucial for species propagation. Juvenile hormone, ecdysteroids, and insulin signaling contribute to oocyte maturation, and nutrient sensing during ovarian development⁴. Growing evidence shows that gut bacteria can maintain these signaling pathways^{5,6}. In *Drosophila melanogaster*, gut microbiome changes can reshape host gene expression^{7,8}; specifically, the downregulation of *Aldehyde dehydrogenase* caused by

the absence of gut *Acetobacter* species is sufficient to suppress oogenesis². Germ-free *D. melanogaster* also exhibits ovarian and even systemic metabolism disorders⁹. Under amino acid-deficient diets, the contribution of gut bacteria to fecundity becomes particularly pronounced in *Bactrocera oleae*¹⁰. Importantly, reproductive defects resulting from gut bacteria deletion can be rescued through specific gut bacteria or dietary supplementation^{1,2,9}.”

I suggest improving the transition between the third and fourth paragraphs. The molecular mechanisms by which gut symbionts regulate insect reproduction are not well understood, and ubiquitination could be one of them.

Answer: Thank you for the suggestion. We have improved it as following: “Although the metabolic and nutritional roles of gut bacteria are closely linked to the reproductive health of insect host, the underlying molecular mechanisms remain largely unknown. Among the potential regulatory pathways, ubiquitination may also play a role” (Line 62-64).

Materials and Methods

L 399: Please provide more information about *Bactrocera dorsalis*, particularly regarding the microbial composition of its gut. Additionally, I wonder whether the flies carry *Wolbachia*? If so, where are they located, and what is their impact on the fly? Have you checked whether eliminating gut bacteria also eliminates this bacterium? Do you think the bacterium could affect your study?

Answer: Thank you for your suggestion. We have provided more information about *Bactrocera dorsalis* gut microbiota in *Introduction*: “*B. dorsalis* has a stable and complex gut bacterial community, with Enterobacteriaceae as the predominant family. *Enterobacter* and *Klebsiella* are the most abundant genera¹¹. Several culturable species, such as *Enterobacter cloacae*, *Klebsiella michiganensis*, *Klebsiella oxytoca*, and *Citrobacter freundii*, have been isolated from the gut and shown to facilitate larval development and contribute to the adaptation of *B. dorsalis* to the environmental stress¹²⁻¹⁴” (New Lines 77-83).

Thank you again for these interesting questions. The influence of *Wolbachia* on insect reproduction has always been of great concern. A previous study screening 1,500 individuals of *B. dorsalis* from different regions in China found that only 19 individuals were infected with *Wolbachia*¹⁵. Although our previous 16S rDNA sequencing did not detect *Wolbachia* in the gut of lab-reared population¹³, we cannot exclude the possibility of infections in other tissues or at different developmental stages. Therefore, we collected first-, second-, and third-instar larvae, newly eclosed adults, as well as head, abdomen, gut, fat body, ovary, and testis, and extracted their DNA. *Wolbachia* infections were screened by PCR using the *wsp* primers: *wsp* 81F (5'-TGGTCCAATAAGTGATGAAGAAAC-3') and *wsp* 691R (5'-AAAATTAAACGCTACTCCA-3')¹⁶. As shown in the Figure. R4 (below), no *Wolbachia* was detected in any of these tissues or developmental stages of *B. dorsalis*. These results indicate that the *B. dorsalis* population used in this study does not harbor *Wolbachia*.

Wolbachia can typically be eliminated by tetracycline or rifampicin. If our flies were harboring *Wolbachia*, the mixed antibiotics (penicillin and streptomycin) used in our study might not have been effective in clearing this bacterium.

There is an enormous diversity of *Wolbachia* strains in nature, induce a diversity of phenotypes on numerous invertebrate host species. They can cause cytoplasmic incompatibility, male killing. In the wasp *Asobara tabida*, *Wolbachia* is required for oogenesis¹⁷. Some

Wolbachia infection events even have no apparent phenotypic effects on hosts¹⁸. Therefore, whether *Wolbachia* infection affects reproduction in *B. dorsalis* remains an interesting question for future investigation.

[editorial note: unpublished data redacted]

L 407: Regarding bacterial elimination and recolonization, could you provide more information on which bacteria were reintroduced and how they were reintroduced? Did you introduce all the bacteria from the gut flies directly, or did you cultivate the bacteria and reintroduce them in the same quantity? And when you introduced only the *E. hormaechei* strain, how much did you introduce (CFU)?

Answer: Thank you for your question. We performed two types of re-introduction experiments: re-introduction of total culturable gut bacteria, and re-introduction of five individual bacterial strains, including *E. hormaechei*.

For the re-introduction of total culturable gut bacteria, 10 female guts were dissected and homogenized in 500 μ L sterile PBS. A 100- μ L aliquot of the gut homogenate was inoculated into 100 mL LB medium and cultured at 37 °C for 8 h. The culture was then centrifuged at 4,000 rpm for 10 min, and the bacterial cells were resuspended in sterile diet to a final concentration of OD₆₀₀ = 5 before feeding to ABX females.

For single-bacterium re-introduction, each bacterial strain was supplied with the diet at a concentration of 1×10^9 CFU/mL: bacteria were cultured in LB medium at 37 °C, and samples were collected at different time points to measure optical density (OD). Serial dilutions were plated on LB agar to determine colony-forming units (CFU) and establish the OD–CFU relationship. Based on this, the resuspension volume of bacteria cells was calculated to standardize the supplemented diet to 1×10^9 CFU/mL.

We have added more details in revised manuscript, please see Lines 492-503.

L 424: Regarding the assessment of gut bacterial load, please specify which 16S rRNA primer pairs you used. Are they universal primers or specific to each bacterium present in this insect?

Answer: Thank you for your question. We used universal bacterial 16S rRNA primers to assess gut bacterial load after antibiotic feeding and re-introduction of total culturable bacteria. The primer sequences were as follows: forward, 5'-ACTCCTACGGGAGGCAGCAG-3', and reverse, 5'-TACCGCGGCTGCTGG-3'. We have revised the manuscript to include the primer information in Line 516.

L 451: Further clarification is needed regarding the study of ovarian development stages. When did you take the measurements (female age)? And were they identical for each experiment?

Answer: Thank you for pointing this out. Ovarian development was assessed in females at 8 days post-eclosion, and this age was consistent across all experiments. This information has now been added to revised manuscript, as shown in Line 573.

Please add descriptive titles for the tables in the supplementary data. Furthermore, additional information on primer pairs needs to be added to Table S2 in the supplementary materials.

Answer: Thank you for your suggestion. Supplementary Table 1 is now entitled “Overlapping

ubiquitinated proteins and sites that are hypo-ubiquitinated and upregulated in ABX vs Ctrl, and hyper-ubiquitinated and downregulated in EH vs ABX (related to Fig. 4e)”.

Supplementary Table 2 is now presented as a separate file with updated descriptive title: “List of primers and siRNA sequences designed for qRT-PCR, gene cloning, and RNA interference experiments”. In addition, we have added further details in the primer list, including the corresponding gene accession numbers, the specific applications of each primer pair, and the target regions of the siRNA sequences.

Results

L 111: Have you tried adding all the metabolites together? Mixing all these metabolites could complete the ovarian development.

Answer: Thank you for this insightful suggestion. We have supplemented all the six metabolites and result showed that it did not fully restore ovarian development ($p = 0.042$), as shown in Figure. R5 (below). The influence of gut microbiota on ovarian physiology involves a complex regulatory mechanism. Our metabolomic analysis of microbiota-depleted females showed that, in addition to the downregulation of certain metabolites, some metabolites were upregulated. Interestingly, other researchers in our laboratory have found that increases in certain metabolites in gut bacteria-depleted females also negatively affect ovarian development (unpublished data). Therefore, it is reasonable that nicotinic acid supplementation in our study, and even the mixture of all these metabolites, only partially rescued ovarian development.

[editorial note: unpublished data redacted]

L 114: Please explain why you tested for vitellogenin deposition.

Answer: Thank you for your question. (This sentence is now on New Line 123) Vitellogenin deposition was measured to reflect ovarian development. In most insects, vitellogenesis is a central event of female reproduction¹⁹, which serves as the primary source of nutrients, including proteins and lipids, for developing oocytes²⁰. In many studies, vitellogenin expression and protein deposition have been used as indicators for evaluating insect reproductive health²¹⁻²³. Therefore, we tested vitellogenin.

LL 132-136: Please split this sentence into two and add that NA is undetectable in hemolymph and ovaries.

Answer: Thank you for pointing this out. Since this experiment aimed to examine whether $\Delta PncA$ *E. hormaechei* can still secrete NA, we revised the manuscript as follows (New Lines 156-160):

“To further investigate the role of NA, we knocked out the *nicotinamidase/pyrazinamidase* gene ($\Delta PncA$), a key gene in the NA synthesis pathway (Supplementary Fig. 4b). NA secretion became undetectable in the bacterial culture (Supplementary Fig. 4c), whereas *PncA* depletion did not affect the growth of *E. hormaechei* in either nutrient-rich or minimal media, nor its gut colonization (Supplementary Fig. 4d, e).”

L 167: Please explain why you checked mitochondrial oxidative phosphorylation.

Answer: Thanks for your question. Mitochondrial oxidative phosphorylation (OXPHOS) produces ATP through electron transfer and the generation of a transmembrane proton gradient. NAD(H) is a critical electron donor in this process, and changes in its levels affect electron

transport chain activity and ATP synthesis²⁴. The capacity of OXPHOS is a key indicator of mitochondrial function, particularly its ability to produce ATP²⁵. Our previous experiments in this study showed that gut microbiota-derived NA influences NAD(H) levels, so we measured OXPHOS activity to assess whether microbiota-derived NA impact mitochondrial energy metabolism. We have explained this: “Thus, we investigated the impact of NA on mitochondrial oxidative phosphorylation (OXPHOS), the main pathway for cellular energy production” (New Line 197).

LL 168-169: I disagree. The results show the opposite. Perhaps this is a color issue?

Answer: We sincerely thank the reviewer for the careful observation. These sentences are now on New Line 199-200. The issue was indeed a color labeling error, and we have now corrected it by replacing the incorrect panel in Supplementary Fig. 5.

L 200: Please explain why you silenced the expression of Uba 1.

Answer: Thank you for your question. Uba1 (ubiquitin-like modifier activating enzyme 1) functions as the primary E1 enzyme responsible for initiating ubiquitin activation, which is the first step in the ubiquitin–proteasome system. It has been reported that inhibition of Uba1 activity reduces the ubiquitination level. Therefore, we silenced *Uba1* to reduce overall ubiquitination, thereby allowing us to assess whether decreased ubiquitination will affect ovarian development.

To avoid confusion in the Results section, we have added a clearer description of Uba1 in the revised manuscript: “To investigate whether reduced ubiquitination is associated with the negative impact on ovarian development in ABX females, we silenced *Uba1*, the major E1 enzyme responsible for initiating ubiquitination, by dsRNA-mediated RNA interference (RNAi)” (New Lines 232-234).

L 276-278: Can you explain why the control in Supplementary Figure 11 is similar to ABX+dsGFP? Furthermore, I believe that Figure 11 should be included in Figure 5.

Answer: Thank you for your question. In the revised manuscript, Supplementary Figure 11 has been renumbered as Supplementary Figure 12. In Supplementary Figure 12a, we measured the *Lolal* mRNA levels after *Lolal* knockdown in ABX females. As our result (Lines 296-297), depletion of gut microbiota affects *Lolal* post-translational modification rather than its transcription. Therefore, there is no significant difference in *Lolal* mRNA levels between the control and ABX+dsGFP groups.

In fact, Supplementary Figure 12 serves as a validation of RNAi efficiency and provides supporting information for Figure 5. For this reason, we included it in the Supplementary Figures rather than in the main figure panel.

L 282: Please explain why there is no effect on the length of mature eggs.

Answer: Thank you for your question. This sentence is now on New Line 335. As discussed in the manuscript (Line 461): “In *Lolal* and *dpp* knockdown normal females, the reduced egg length likely due to impaired transfer of nurse cell contents to the oocyte caused by an irregular F-actin network^{26,27}”. In ABX females, *Lolal* protein and *dpp* mRNA levels were elevated compared with the control group (manuscript Fig. 4i and 5o), which may not disrupt actin network. Moreover, *Lolal* or *dpp* knockdown in ABX females did not reduce their levels below

those of control, as shown in manuscript Supplementary Fig. 12b, c. Therefore, the length of mature eggs was not affected.

L 290: Why are Figure 5a and Figure S12a different?

Answer: Thank you for your question. In the revised manuscript, Figure S12a has been renumbered as Figure S13a. In Figure 5a, we knocked down *Lolal* and measured the mRNA levels of *Lolal* and *dpp*. In Figure S13a, we knocked down *dpp* and examined the expression levels of *dpp* and *Lolal*. Therefore, the two figures represent different experimental conditions.

Figures

Figure 1: I find the figures too small and difficult to read. I would recommend changing it to portrait format and enlarging all the figures it contains.

Answer: Thanks for your suggestion. Figure 1 has been changed to portrait format, and all panels have been enlarged to improve readability.

Figure 2: Figures 2d and j must be adjacent because they are identical.

Answer: Thanks for your suggestion. We have changed it in right order and revised the manuscript accordingly.

Figure 4: Figures should be positioned in order.

Answer: Thanks for your suggestion. We have changed it in right order.

Remarks on code availability

Raw data and scripts are not available.

Proteomic and metabolomic data have to be deposited in a public data repository.

Answer: Thanks for your suggestion. No custom scripts were used in this study; all data analyses were performed using publicly available software and online tools as described in the Materials and Methods section.

The proteomic, metabolomic and 16S rDNA sequencing raw data have been deposited in public repository and the accession numbers have been provided in revised manuscript (please see New Lines 814-819).

Reviewer #2 (Remarks to the Author):

This study uses *Bactrocera dorsalis* as a model system to investigate how gut symbiotic microbes influence host insect reproduction (ovarian development), not only from the perspective of nutritional metabolism but also in terms of gene expression regulation. The authors first performed RNA-seq analyses and discovered that niacin (vitamin B3), a metabolite produced by gut bacteria, may play a crucial role. The authors then demonstrated that microbial NA indeed affects ovarian development by reinfesting antibiotic-treated flies with gut bacteria, as well as by infecting them with bacterial mutants deficient in NA biosynthetic genes.

Beyond such experimental validation, the authors further examined the mechanism by which vitamin B3 regulates ovarian development. Using detailed biochemical and reverse genetic approaches (RNAi), they showed that reduced NA levels lead to decreased mitochondrial activity, which in turn reduces ubiquitination of Lolal, a regulatory factor of the growth factor decapentaplegic (dpp). As a result, dpp expression increases, ultimately suppressing ovarian development in the insect.

Although symbiotic bacteria are known to contribute significantly to ovarian development and reproduction in insects, the underlying mechanisms (or causal relationships) have remained unclear. This study partly elucidates how bacterial metabolites influence insect gene regulation and lead to reduced ovarian development. Although, the manuscript is generally well written and well organized, I still have several major and minor concerns, as outlined below.

Answer: We sincerely thank you for these encouraging comments on our work. We have carefully considered all the concerns and have revised the manuscript accordingly. Specifically, we have added more information on the gut microbiota background and statistical analysis, and provided additional data supporting our conclusion.

Major Concerns

1. Gut microbiota characterization

No information is provided regarding the gut microbiota of conventionally reared *Bactrocera dorsalis*. Previous studies have shown that the composition and diversity of gut microbiota can be influenced by both host strain and food resources (Tian et al. 2023; *Frontiers in Microbiology*). Including background data on the gut microbiota—such as NGS results—is critical for interpreting and validating the downstream experiments.

Answer: We appreciate the reviewer's professional question. We have added more background information about the *B. dorsalis* gut microbiota in the Introduction section: "*B. dorsalis* has a stable and complex gut bacterial community, with Enterobacteriaceae as the predominant family. *Enterobacter* and *Klebsiella* are the most abundant genera¹¹. Several culturable species, such as *Enterobacter cloacae*, *Klebsiella michiganensis*, *Klebsiella oxytoca*, and *Citrobacter freundii*, have been isolated from the gut and shown to facilitate larval development and contribute to the adaptation of *B. dorsalis* to the environmental stress¹²⁻¹⁴" (New Lines 77-83).

To investigate the gut microbial composition of *B. dorsalis* females, we performed gut bacterial 16S rDNA sequencing for conventionally reared females (Figure R6, below). Result showed that 97.8% of the gut microbiota belonged to the family Enterobacteriaceae, with *Enterobacter* and *Klebsiella* being the predominant genera, which is consistent with our previous study²⁸.

The new data have been incorporated into the revised manuscript (Please see Line 93-96 and the New Figures: Supplementary Fig. 1a and 1b). Thanks again for your suggestion.

Figure R6. Relative abundance of gut bacteria in conventionally reared females (Ctrl) at the family level (a) and genus level (b).

Tian et al. and some other studies²⁹⁻³¹ have found the dynamic changes in gut microbiota among different host plants and highlighted its crucial role in enabling insects to adapt to different hosts and environmental conditions. In our previous study, we characterized the gut microbial composition of *B. dorsalis* from laboratory-reared, sterile sugar-fed, and field-collected populations²⁸. It also found that different environmental conditions and food supply could influence the diversity of the harbored bacterial communities and increase community variations²⁸. However, some gut bacteria were stable in these populations, with Enterobacteriaceae being the dominant family, and *Enterobacter*, *Klebsiella*, and *Citrobacter* as the major genera. The *B. dorsalis* population used in this study has been maintained in the laboratory for several years. The larvae were continuously reared on artificial diet containing banana, and the adults were fed a mixture of yeast and sucrose. Indeed, it has reported that the establishment and maintenance of the gut microbiome mainly rely on the ingestion of environmental bacteria in *Drosophila*³². Therefore, it is highly likely that the flies harbor a stable gut bacterial community, under these controlled environmental conditions.

2. Selection of bacterial strains

How diverse are the microbes that colonize the midgut of *B. dorsalis*? And how were the five bacterial strains used in this study selected? Were only these five strains recovered through culturing (L121–126)? Providing a rationale based on the actual composition of the gut microbiota is necessary to justify the experimental design.

Answer: Thanks for your suggestion. The gut microbial composition of *B. dorsalis* females is shown in Figure R6, above. In addition, our previous study revealed that the gut microbiota of *B. dorsalis* is dominated by Gammaproteobacteria, while Actinobacteria and Firmicutes accounted for more than 10% of the community. Smaller proportions of Flavobacteria, Deltaproteobacteria, Bacteroidetes, and Alphaproteobacteria were also observed²⁸. In males gut, Enterobacteriaceae accounted for 69.35%, Enterococcaceae for 9.61%, Bacillaceae for more than 6% (Figure R7a, below), and 95.41% of the cultivable bacteria belonged to Enterococcaceae¹³, as shown in Figure R7 below (cited from Cai *et al.*, 2018, *Evolutionary applications*, Figure 6).

The five bacterial strains used in our study were selected based on our previous study¹, in which cultivable bacteria were isolated from the guts of adult females. It showed that a total of 10 cultivable gut bacterial species were isolated, including *Klebsiella aerogenes*, *Enterobacter hormaechei*, *Providencia vermicola*, *Providencia alcalifaciens*, *Klebsiella quasipneumoniae*, *Klebsiella variicola*, *Providencia rettgeri*, *Klebsiella pneumoniae*, *Proteus mirabilis*, *Citrobacter farmeri*. Among these, the relative abundances of *K. aerogenes*, *E. hormaechei*, *P. vermicola*, *P. alcalifaciens*, and *K. quasipneumoniae* were higher, as shown in Figure R8 (below) (Zhang *et al.*, 2025, *Cell Rep*, Figure S1F). We further found that they possess complete nicotinic acid biosynthetic pathways by genomic analysis. Therefore, these five strains were selected for use in this study.

We have revised the manuscript to provide a rationale based on the actual composition of the gut microbiota: “Our previous study identified *Klebsiella aerogenes*, *Enterobacter hormaechei*, *Providencia vermicola*, *Providencia alcalifaciens*, and *Klebsiella quasipneumoniae* as dominant culturable bacterial species in the female gut¹. To determine potential NA-producing bacterial providers, we assessed the NA production capacity of these isolates using LC-MS/MS” (New Lines 138-142).

[editorial note: 3rd party material redacted]

Figure R7 (Cai *et al.*, 2018, *Evolutionary applications*, Figure 6). Diversity of gut bacteria in *B. dorsalis*.

[editorial note: 3rd party material redacted]

Figure R8 (Zhang *et al.*, 2025, *Cell Rep*, Figure S1F). The abundance of culturable bacteria in the female gut.

3. Assessment of reproductive ability

To evaluate female reproduction, the authors measured ovarian development, mean egg length, and total egg number. Did gut microbiota influence only ovarian development without affecting the timing of oviposition? Enhanced ovarian development could plausibly accelerate oviposition. Clarification would strengthen the conclusions.

Answer: We thank the reviewer for raising this important point. To clarify this point, we examined oviposition time of 11-day-old *B. dorsalis* females from the Ctrl, ABX, ABX+EH, and ABX+NA groups. On the first day after mating, oviposition attempt was observed in all females of these groups, but not all females lay eggs (Figure R9a, below). We further analyzed the timing of first oviposition (i.e., the first appearance of egg output) and found that ABX females exhibited a 1.3-day delay on average compared with Ctrl females, whereas this difference was not statistically significant (Figure R9b, below). These results indicate that gut microbiota did not significantly affect the timing of oviposition under our experimental conditions. Instead, the influence of gut microbiota on reproduction is primarily mediated through ovarian development, as ABX females showed oviposition attempt but lacked sufficient mature eggs in ovary to support actual egg laying (Figure R9c, below). A study from the Jianping Chen lab on *R. pedestris*³³ also suggested that the lower fecundity in gut microbial-depleted females are due to undeveloped ovaries.

The new data about oviposition time have been incorporated into the revised manuscript (Please see Line 127 and the New Figure: Supplementary Fig. 3e).

Figure R9. Analysis of reproductive performance and time to first reproduction. (a) The number of eggs laid during the first oviposition attempt. $n = 15$ for Ctrl, ABX, ABX+ EH, and ABX+ NA; $n = 13$ for ABX+ Δ PncA EH. (b) The first time to reproduction. $n = 15$ for each. [editorial note: unpublished data redacted] Kruskal–Wallis test followed by Dunn’s multiple comparisons test; for (a), $p < 0.001$; [editorial note: unpublished data redacted]; All variables with different letters are significantly different; ns, non-significant.

4. Interpretation of dsRNA experiments (Fig. 5)

The authors tested that bacteria-derived NA modulates the *Lolal*–*dpp* signaling pathway to influence ovary development. However, in the RNAi experiments, it is still difficult to conclude that silencing *Lolal* or *dpp* rescues ovary development. In fact, ovarian size and maturation do not appear to be restored (Fig. 5e, f), and overall fecundity is significantly reduced. Although the authors provide some explanations (L290–303), these data do not convincingly support the claim that Fig. 5 provides mechanistic evidence for microbial regulation of ovary development. *Answer:* Thanks for your suggestion. Fig. 5e, f has been renumbered as Fig. 5j, k in new manuscript. We conducted a statistical analysis of ovarian development and fecundity data in the RNAi experiments. The result showed that, compared with the ABX + dsGFP group, both *Lolal* and *dpp* RNAi in ABX females significantly increased the proportion of grade V ovaries and enhanced egg production (Figure R.10a and 10b). These results indicate that *Lolal* and *dpp* knockdown partially rescued the reproductive defects caused by the depletion of gut bacteria, rather than having no effect at all.

Given the RNAi experiments in ABX females was to investigate whether elevated *Lolal* protein in ABX female contributes to impaired female reproduction, we employed a more direct approach to further support our conclusion. To do so, we expressed and purified recombinant *Lolal*-His protein and then injected it into two-day-old normal females, with GST-His serving as a control (Figure. R10c). We found that *Lolal*-His injection markedly promoted *dpp* expression, inhibited ovarian development, and reduced egg production (Figure. R10d-f). These inhibitory effects on female reproduction after *Lolal* overexpression further supports our conclusion that *Lolal* mediates the influence of gut bacteria on ovarian development.

The new data have been incorporated into the revised manuscript (Please see the New Lines 315-323 and Figures 5d-5i). Thanks again for this constructive suggestions.

Figure R10. **Lolal-dpp acts as a mediator in gut bacterial regulation of reproduction.** (a, b) development grades (a) and egg production (b) of Ctrl, ABX, *Lolal* RNAi and *dpp* RNAi ABX females. $n > 40$ per group for (a), $n = 8$ for (b). (c) SDS-PAGE and western blot analysis for recombinant Lolal-His and GST-His. (d, e) Representative images of ovaries (d) and the development grades (e) in females injected with Lolal-His and GST-His. $n > 60$ per group. (f, g) Relative expression of *dpp* (f) and egg production (g) in females injected with Lolal-His and GST-His. $n = 3$ for (f); $n = 8$ for (g). (a, e) Pearson's Chi-square test followed by post-hoc pairwise comparisons with Bonferroni correction. (b) one-way ANOVA followed by Tukey's multiple comparisons test, $p < 0.0001$. (f, g) unpaired *t*-test. All variables with different letters are significantly different; *** $p < 0.001$.

5. Statistical analysis

Statistical details are insufficient. Throughout the manuscript, values such as t-values, F-values, and degrees of freedom are missing. For two-way ANOVA analyses, the interaction term between factors is important for assessing treatment effects on fecundity, yet this is not described. In addition, there is no explanation of how multiple comparisons were performed. In figures such as Fig. 1g, significance is indicated with letters, but the statistical test and significance level are not stated. The reporting of statistical analyses and figure legends is inadequate. If statistical methods beyond ANOVA were used, they must be described explicitly. **Answer:** We appreciate the reviewer for pointing this out. We have updated the results throughout the manuscript to include the statistical test names, p-values, and test values. Specifically, we report $t(df)$ values for unpaired *t*-test, $F(df)$ values for one-way ANOVA, $\chi^2(df)$ for Pearson's Chi-square test, H values for Kruskal-Wallis test. Please see the revised manuscript.

In the revised manuscript, two-way ANOVA was used to analyze the differences in *Lolal* degradation rates after drug treatment. We examined the effects of each factor as well as the interaction between factors. The results have been updated accordingly in Lines 287, 289:

“Treatment with the proteasome inhibitor MG132 (carbobenzoxy-Leu-Leu-leucine) delayed degradation of *Lolal* protein (two-factor ANOVA, $p_{\text{treatment}} < 0.0001$, $p_{\text{time}} < 0.0001$, $p_{\text{interaction}} < 0.0001$; Fig. 4h), while treatment with the lysosome inhibitor chloroquine (CQ) did not prevent degradation (two-way ANOVA, $p_{\text{treatment}} = 0.078$, $p_{\text{time}} < 0.0001$, $p_{\text{interaction}} = 0.069$; Supplementary Fig. 10b)”.

We have updated the Figure legend to specify the statistical tests used for each panel: the methods for multiple comparisons and the significance levels. Specifically, one-way ANOVA was followed by Tukey’s multiple comparison test, and the Kruskal–Wallis test was followed by Dunn’s multiple comparison test. The Materials and Methods section now includes an overview of the statistical analyses applied throughout the manuscript, as shown in Lines 797–812.

Minor Comments

•L19 and L26: These two sentences are identical. In the abstract, this is redundant.

Answer: We sincerely thank the reviewer for highlighting this issue in the abstract. The old Line 19 is now on New Line 18. Old Line 26 has been clarified and now reads:

“Ubiquitinome analysis further revealed that gut bacteria enhance *Lolal* ubiquitination and promote its degradation” (New Lines 24–25).

•L44: “B” vitamins? Please clarify.

Answer: Thank you for pointing this out, it is indeed B vitamins. We have clarified it in revised manuscript (Line 42).

•L52: The phrase “genetic advantage” is unclear.

Answer: Thank you for pointing this out. The term *genetic advantage* refers to microbial genes that complement host functions. To clarify this, we have revised the sentence in the manuscript as follows (Lines 48–50):

“These host-complementary genetic features expand the functional diversity of microbiota and establish the inseparable host-microbiota relationship.”

•L100: Consider rephrasing as “a key metabolite.”

Answer: Thanks for your suggestion. Insect hemolymph carries nutrients from the gut or fat body to other tissues. Therefore, “Hemolymph serves as a key medium for the systemic transport of metabolites” maybe better. (New Line 108)

•L102: The authors describe “metabolomic analysis,” but the experiment appears to be transcriptomic (RNA-seq).

Answer: We appreciate the reviewer’s comment. We would like to clarify that the analyses described in the manuscript are based on metabolomic data, as correctly stated in the manuscript (New Line 110). In addition, we have carefully reviewed other parts of the manuscript to ensure that no confusing wording remains.

•L135: The authors state bacterial growth is unaffected. Does this apply only in nutrient-rich media? Wouldn’t growth be impaired in minimal medium?

Answer: We appreciate the reviewer’s professional suggestions. We found that deletion of *PncA* did not affect the growth rate of the bacteria in nutrient-rich LB media. Moreover, there was no difference in the growth rate of *E. hormaechei* and $\Delta PncA$ *E. hormaechei* in minimal medium (M9 medium) as shown in Figure. R11 (below). This can be explained as follows: The catalytic product of *PncA*, nicotinic acid (NA), acts as a precursor substance and is converted through the Preiss-Handler pathway into bioactive NAD(H). However, unlike insects, bacteria can utilize multiple precursor substrates through alternative biosynthetic pathways to synthesize NAD(H). That is why the absence of *PncA* has no effect on the growth rate of *E. hormaechei*.

The new data have been updated in Supplementary Fig. 4d and incorporated into the manuscript Lines 159-160.

Figure R11. *E. hormaechei* and $\Delta PncA$ *E. hormaechei* growth rates in LB medium and M9 medium.

•L184: The claim that these results indicate reduced UPS activity seems logically weak. A supporting reference is needed.

Answer: We sincerely thank the reviewer for pointing this out. The proteasome has known as the central proteolytic complex responsible for degrading most ubiquitinated proteins with ubiquitin chains serving as a signal for recognition^{34,35}. To clarify this we have added a reference and revised the text: “Mechanistically, the proteasome recognizes ubiquitin chains and degrades polyubiquitinated proteins, which plays a central role in protein homeostasis³⁴” (New Line 214).

•L232: There is a logical gap. What about the other 26 genes—are they irrelevant?

Answer: Thanks for your question. Among the 27 proteins, 19 proteins including *Lolal* were enriched in developmental processes according to GO analysis. We further found that *Lolal* had the highest expression level in the ovary among this set (Figure R12, below). These findings imply that *Lolal* may mediate gut bacteria-induced regulation of ovarian development.

The new data have been incorporated into the revised manuscript (Please see Lines 269-272 and the New Figure: Supplementary Fig. 9c).

Figure R12. Relative expression levels of 19 genes in the ovary. n = 4 for each. Unpaired *t*-test; **p* < 0.05.

•L262: The subject would read more naturally as “gut bacteria” rather than “Lolal.”

Answer: Thanks for your suggestion. We have revised this sentence to “To determine whether gut bacteria regulate *B. dorsalis* reproductive development via Lolal-dpp pathway” in Line 304.

•L294: Is 1.09 mm the mean value? If so, the standard deviation should also be included.

Answer: Yes, we have corrected it as “1.09 ± 0.05 mm” (Line 346). Thanks for your suggestion.

•L355: Should read “Vitamin B3 deficiency.”

Answer: Thank you for pointing this out. We have corrected it in Line 427.

•L372: Define “GSC.”

Answer: Thanks for your suggestion. We have added the definition in Line 439: “germline stem cell (GSC)”.

•L436: Yeast extract typically contains vitamin B3. If it was used as a food source, why does deficiency occur?

Answer: Thanks for your question. In our experiments, the food for *B. dorsalis* was of poor yeast content, only 1.5% (sucrose, yeast extract, and water in a 6:1:60 ratio). This diet represents nutritional restriction for the insects³⁶, leading to limited availability of vitamin B3. Under this condition, the flies rely on gut microbiota to provide additional vitamin B3. In *Drosophila*, commensal bacteria can alleviate the adverse effects of a nutritionally poor diet on growth and reproduction^{10,37}. The reproductive performance of *Bactrocera oleae* also relies more heavily on gut symbionts when maintained on a nutrient-deficient diet³⁸. Our previous work further revealed that the growth of *B. dorsalis* larvae requires the participation of gut microbiota to supply vitamin B6 under low-yeast dietary conditions¹².

This point has also been discussed in our revised Discussion section (Lines 394-397):

“In natural environments, insects often face fluctuating or limited nutrient availability. They have evolved nutritional symbiosis with microbial partners that provide essential nutrients, particularly for amino acids and vitamins that insects cannot synthesize de novo or obtain sufficiently from their diet.”

•L459: Please clarify “120 minutes.”

Answer: We thank the reviewer for pointing this out. In the manuscript, “120” refers to the

number of passages of the *B. dorsalis* embryonic cell line, not time. The sentence has been clarified: “The *B. dorsalis* embryonic cell line was established from insect eggs and has been stably passaged more than 120 times” (New Line 533).

•L543: Information on primer sets used for dsRNA synthesis should be provided.

Answer: We totally agree with that. The primer sequences used for dsRNA synthesis are provided in Supplementary Table 2.

•L565: Please specify the plasmid into which Lolal was cloned.

Answer: Thanks for your suggestion. The Lolal coding sequence was cloned into the pBac-IE1 plasmid. It has been updated in revised manuscript Line 655.

•L580: The antibody production method is unclear. If a commercial source was used, please state the supplier.

Answer: Thanks for your suggestion. The antibody against Lolal was custom-produced by AtaGenix (Wuhan, China). We have revised the *Materials and Methods* section as follows (Lines 669-674):

“The recombinant protein was purified with Ni NTA Beads 6FF (SMART Life Sciences, China) and used by AtaGenix (Wuhan, China) to immunize rabbits for polyclonal antibody production. Rabbits were initially injected subcutaneously with antigen emulsified in complete Freund’s adjuvant, followed by two booster injections with incomplete Freund’s adjuvant at 1-week intervals. Serum samples were collected 10 days after the final boost, and the resulting antisera were affinity-purified before being applied in immunoblotting.”

•Fig. 1g, i, j: The y-axis label (“NA related content”) should include units.

Answer: Thanks for your suggestion. The NA levels in these figures were presented as relative content rather than absolute concentration. Therefore, no specific units are applicable. To avoid confusion about the term “content”, we have revised the y-axis label to “Relative NA level” in Fig. 1h, j, and k.

Reviewer #3 (Remarks to the Author):

Qiao and colleagues have investigated the relationship between the gut microbiota, ovarian development and consequent fertility in the oriental fruit fly, *Bactrocera dorsalis*. They showed that gut symbionts, particularly *Enterobacter hormaechei*, produce nicotinic acid which improves mitochondrial function and promotes the ATP-mediated ubiquitination and consequent degradation by the proteasome pathway of a transcription factor, Lolal, regulating the expression of *dpp* in ovaries.

This is a well written manuscript and a thorough investigation of molecular interactions between gut symbionts and ovarian development, using an impressive and complementary methods palette to assemble a coherent mechanistic picture. I don't have technical concerns. The experiments seem to have been well-performed, are consequential and complementary, and together support the mechanistic claims. I am excited about this study and have only a few general recommendations to add an evolutionary angle to the interpretation of these results.

The manuscript would benefit from discussing why ovarian development and fertility should depend so strongly on gut microbial signals. What is the evolutionary rationale for such dependence, given that imperfect transmission or acquisition of gut symbionts could severely reduce reproduction? Could the authors add some introduction and discussion on the evolutionary implications of their findings and explain whether such connection makes sense from an evolutionary perspective? The study focuses on ovarian development and didn't touch upon other potential developmental effects of these NA-Lolal-*dpp* interactions. Are these effects specific to reproduction or by-products of nutritional deficiencies hampering development more in general? Given the loss of fertility, these insects would not do well if a specific set of symbionts is not faithfully acquired or is subsequently disrupted. While the introduction already states that *Bactrocera dorsalis* flies host a stable and complex microbial community dominated by Enterobacteriaceae, a bit more background of this would be helpful. Are *Enterobacter hormaechei*, or other symbionts capable of producing NA, always present within the gut of individual flies in nature? How are they acquired? Are they vertically transmitted? Can they be lost?

Answer: We sincerely thank you for the encouraging evaluation of our work. Your insightful comments have greatly enriched our manuscript. In particular, your suggestion to interpret the findings from an evolutionary perspective has encouraged us to reflect more deeply on this aspect. The question regarding transmission route of *E. hormaechei* is also valuable and aligns closely with our future research directions. For clarity, we have divided your comments into specific points and provided detailed Answers to each below.

1. The manuscript would benefit from discussing why ovarian development and fertility should depend so strongly on gut microbial signals. What is the evolutionary rationale for such dependence, given that imperfect transmission or acquisition of gut symbionts could severely reduce reproduction? Could the authors add some introduction and discussion on the evolutionary implications of their findings and explain whether such connection makes sense from an evolutionary perspective?

Answer: Thanks for your suggestion. Many studies have revealed a close relationship between insect reproduction and their symbionts. Symbionts can influence host reproductive physiology by modulating insulin and TOR signaling pathways. Developing oocytes rely on gut microbiota

to supply essential nutrients, particularly under nutrient-deficient conditions. These microbial metabolites function not only as nutritional sources but also as precursors of active coenzymes that drive metabolic reactions. Moreover, some metabolites can modulate epigenetic states in the reproductive system, thereby influencing reproductive processes. We have expanded these in New Lines 51–61 and 422–437.

For evolutionary implications, in the Introduction, our manuscript has emphasized the genomic-level complementarity between microbiota and their hosts during coevolution. Please see Lines 37-50 in revised manuscript.

We have revised and expanded the Discussion section to further elaborate on the close association between insect reproduction and their microbial symbionts, which focus on the mutualistic relationship between symbiotic bacteria and their insect hosts under environmental pressures, as well as host-driven selection, together forming a mutually beneficial symbiosis. Please see Lines 393-421 in revised manuscript.

2. The study focuses on ovarian development and didn't touch upon other potential developmental effects of these NA-*Lolal-dpp* interactions. Are these effects specific to reproduction or by-products of nutritional deficiencies hampering development more in general? Given the loss of fertility, these insects would not do well if a specific set of symbionts is not faithfully acquired or is subsequently disrupted.

Answer: We appreciate the reviewer's insightful comments. Based on our current data, we consider these effects to be specific to reproduction: In addition to ovarian development, we also examined the potential effects of NA-*Lolal-dpp* interaction on lifespan and egg hatching rate. Antibiotic treatment and gut bacterial reinfection did not affect fly survivorship (Figure R13a, below). Moreover, the NA-*Lolal-dpp* pathway also influences egg hatching rate (Figure R13b and c, below). Previous studies have shown that maternally derived *Lolal* is involved in dorsal-ventral axis formation³⁹, which may explain the embryonic developmental failure observed in ABX flies. We also attempted to generate *Lolal* mutants using the CRISPR-Cas9 system to further investigate *Lolal* function and found that *Lolal*-depleted individuals (G0 generation) exhibited normal hatching rates, whereas most larvae failed to develop to the pupal stage (Figure R13d and e, below). This suggested that zygotic *Lolal* is essential for larval development. Taken together, our findings suggest that the effects of the NA-*Lolal-dpp* are primarily associated with reproductive phenotypes, including ovarian development, egg hatching rate, and larval development. Moreover, disruption of these pathways, independent of gut microbial status, directly impairs reproductive processes rather than being mere secondary effects of general nutrient deficiency.

We have added the new data about survivorship, egg hatching rates in revised manuscript. (Please check the New Figures: Fig. 1g, Fig. 5m, and Supplementary Fig. 2c)

Although depletion of gut microbiota can lead to nutritional deficiency, our findings reveal the specific molecular disruptions that underlie this process. NA is distributed to various tissues and converted into the active coenzyme NADH, which plays essential roles in metabolic processes²⁴. NA supplementation increased *Lolal* ubiquitination and enhanced *dpp* transcription, suggesting a specific role of NA in regulating *Lolal* and *dpp* expression. Moreover, we performed rescue experiments in ABX females by *Lolal* and *dpp* RNAi trying to restore their expression levels to those of the control group. This partially rescued the reproductive defects

in these females, as shown in Figure. 5j, 5k, 5l, 5m (New manuscript). These findings indicate that the *Lolal-dpp* pathway functions downstream of gut bacteria as a regulatory axis. Therefore, the influence of gut microbiota on reproduction cannot be solely attributed to nutrient availability; rather, it involves more sophisticated molecular regulatory mechanisms, among which the *Lolal-dpp* pathway plays a key role.

Figure R13. Other potential developmental effects of NA-*Lolal-dpp*. (a) Survival curves by log-rank (Mantel–Cox) analysis (individual = 80). (b) Egg hatching rates of Ctrl, ABX, and ABX + NA females. (c) Egg hatching rates of Ctrl, ABX, and *Lolal* RNAi, *dpp* RNAi ABX females. [editorial note: unpublished data redacted] (b, c) one-way ANOVA followed by Tukey’s multiple comparisons test, $p < 0.0001$; [editorial note: unpublished data redacted].

3. While the introduction already states that *Bactrocera dorsalis* flies host a stable and complex microbial community dominated by Enterobacteriaceae, a bit more background of this would be helpful.

Answer: Thanks for your suggestion. We have added more background in the *Introduction* section, as shown in New Lines 77-83: “*B. dorsalis* has a stable and complex gut bacterial community, with Enterobacteriaceae as the predominant family. *Enterobacter* and *Klebsiella* are the most abundant genera¹¹. Several culturable species, such as *Enterobacter cloacae*, *Klebsiella michiganensis*, *Klebsiella oxytoca*, and *Citrobacter freundii*, have been isolated from the gut and shown to facilitate larval development and contribute to the adaptation of *B. dorsalis* to the environmental stress¹²⁻¹⁴”.

4. Are *Enterobacter hormaechei*, or other symbionts capable of producing NA, always present within the gut of individual flies in nature? How are they acquired? Are they vertically transmitted? Can they be lost?

Answer: Thank you for your insightful question. In previous study⁴⁰, *E. hormaechei* could be isolated from field-collected *B. dorsalis* populations, along with other bacterial species such as *Enterobacter cloacae*, *Klebsiella oxytoca*, *Pantoea dispersa*, *Enterobacter aerogenes*, *Enterococcus faecalis*, *Bacillus cereus*, and *Bacillus anthracis*. We further analyzed the genomes of these bacteria and found that other bacteria carrying a complete nicotinic acid (NA) biosynthetic pathway are present in natural *B. dorsalis* populations (as shown in Table R2 below). This point has been added in Line 406-407. Similarly, Chu *et al.* reported the presence of *Enterobacter hormaechei* in the field-collected *Spodoptera frugiperda* adults⁶. *E. hormaechei* also was identified as the dominant bacterium in the field-collected *Plutella*

xylostella populations⁴¹. These studies indicate that *E. hormaechei* constitutes part of the gut bacterial community in multiple insect species and can even dominate the gut microbiota in some insects.

We are also interested in exploring how *E. hormaechei* is transmitted within *B. dorsalis*. In our previous study, *E. hormaechei* was not detected in ovary by fluorescence in situ hybridization (FISH)¹. To further examine the possibility of vertical transmission, we fed *B. dorsalis* females with *E. hormaechei* harboring a dsRed-expressing plasmid and collected ovarian tissues, then plated the ovarian homogenates on LB agar to detect dsRed-labeled *E. hormaechei* colonies. However, no dsRed-positive *E. hormaechei* colonies were observed in the LB agar (Figure R14, below). These results suggest that *E. hormaechei* may not be acquired through vertical transmission via the ovary.

We first detected the presence of *E. hormaechei* in the adult gut as early as 2014⁴⁰. Subsequently, other colleagues found that *E. hormaechei* could be isolated from the guts of both males and larvae, and it was not lost even after irradiation treatment¹³. Insects can acquire beneficial gut bacteria from the environment and establish contact with them. *E. hormaechei* is known to be widely distributed in soil and water⁴² and function as a plant bacterial endophyte^{43,44}. Based on these, we suppose that insects are likely to acquire *E. hormaechei* from environmental or dietary sources rather than through vertical transmission. However, the precise origin, transmission routes, and the ecological status of *E. hormaechei* within the gut microbiota across different developmental stages of the insect remain to be further investigated.

[editorial note: unpublished data redacted]

A few minor suggestions:

Some statements in the abstract and introduction are too broad and should be made more conditional. Line 16: not all commensal microbiota shape reproduction and not in every animal, please modify to “..microbiota can play an integral role...”. Similarly, in Line 35: “The insect gut can comprise complex and diverse symbiotic systems”. Line 41: “..this gap can be filled by gut microbiota”. Line 55: “..microbiota can be associated with insect reproductive...”

Answer: Thank you for pointing these out. We have revised them in the corresponding parts of the manuscript (Line 14, Line 33, Line 39). In addition to these points, we also reviewed other sections and made adjustments to avoid overly broad statements (Line 55, Line 393).

Lines 225 and 226: This may be due to my limited knowledge of this specific technique, but I did not fully understand how the 819 and 665 proteins were counted and relate to the numbers given in the scatterplots in Figures 4b and 4c. There is also no explanation in the figure caption about what these numbers (e.g., “1(652)” at the top left of panel b) mean. Could you please improve clarity in this passage?

Answer: Thank you for your question. (Now, these contents are in New Line 261, 263) Figures 4b and 4c show nine-quadrant plots resulting from the combined analysis of the proteome and ubiquitinome, reflecting the relationship between protein abundance changes and ubiquitination changes.

For example, in Figure 4b (ABX/Ctrl): The green points in the top-left quadrant represent proteins with decreased abundance but increased ubiquitination levels (652 proteins); The red points in the top-right quadrant represent proteins with both increased abundance and increased

ubiquitination (602 proteins); The blue points in the bottom-left quadrant represent proteins with both decreased abundance and decreased ubiquitination (424 proteins); The purple points in the bottom-right quadrant represent proteins with increased abundance but decreased ubiquitination (167 proteins).

So, the total of 819 proteins refers to those showing a negative correlation between protein abundance and ubiquitination changes in ABX/Ctrl, which is the sum of 652 (top-left) and 167 (bottom-right) proteins. The 665 proteins correspond to the sum of 49 (top-left) and 616 (bottom-right) proteins.

We have updated the figure legend to clarify the notation (Lines 1070-1072): “Each point represents a protein or K-GG site. The plot is divided into nine quadrants (numbered 1–9), with the number of points in each quadrant indicated in parentheses”.

It is also not completely clear to me how the authors moved from studying proteome-wide ubiquitination and protein abundance to focusing specifically on *Lolal*. I understand that the analysis showed that 27 proteins are both hypo-ubiquitinated and upregulated in the comparison between ABX and ctrl and hyper-ubiquitinated and downregulated in the comparison between EH and ABX. The next step then shows that *Lolal* is among about 20 proteins presented in Supplementary Figure 8b, but I could not understand how the role of these proteins, and only these, in development was determined. A more precise clarification of how this “development” set was identified, its overlap with the 27 proteins above, and the reasoning for focusing on *Lolal* specifically would help here, as this is a key step.

Answer: We appreciate the reviewer for pointing this out. This confusion may have arisen from an unclear description in the previous version. Gene Ontology (GO) enrichment analysis of these 27 proteins indicated that 19 of them were associated with developmental processes. We then analyzed the expression profiles of these 19 proteins in the ovary and found that *Lolal* gene showed the highest expression level (Figure R15, below). Furthermore, tissue expression analysis indicated that *Lolal* expression was significantly higher in the ovary than in other tissues, suggesting that *Lolal* may play an important role in gut microbiota-mediated ovarian development.

The new data have been incorporated into the revised manuscript (Please see Lines 269-272 and the New Figure: Supplementary Fig. 9c, d). Thank you again for pointing this out.

Figure R15. (a, b) Relative expression levels of 19 genes in the ovary (a) and Lolal relative expression levels in tissues (b). (a) $n = 4$ for each; unpaired t -test, $*p < 0.05$. (b) $n = 3$ for each; one-way ANOVA followed by Tukey's multiple comparisons test, $p < 0.0001$.

References

1. Zhang Q, et al. Gut commensal bacteria-derived methionine is required for host reproduction by modulating RNA m6A methylation of the insulin receptor. *Cell Rep.* **44**, 115911 (2025).
2. Elgart M, Stern S, Salton O, Gnainsky Y, Heifetz Y, Soen Y. Impact of gut microbiota on the fly's germ line. *Nat. Commun.* **7**, 11280 (2016).
3. Xie T, Spradling AC. decapentaplegic is essential for the maintenance and division of germline stem cells in the Drosophila ovary. *Cell* **94**, 251-260 (1998).
4. Berg C, Sieber M, Sun J. Finishing the egg. *Genetics* **226**, (2024).
5. Shin SC, et al. Drosophila microbiome modulates host developmental and metabolic homeostasis via insulin signaling. *Science* **334**, 670-674 (2011).
6. Chu B, et al. Gut symbiotic bacteria enhance reproduction in *Spodoptera frugiperda* (J.E. Smith) by regulating juvenile hormone III and 20-hydroxyecdysone pathways. *Microbiome* **13**, 132 (2025).
7. Erkosar B, Defaye A, Bozonnet N, Puthier D, Royet J, Leulier F. Drosophila microbiota modulates host metabolic gene expression via IMD/NF- κ B signaling. *PLoS One* **9**, e94729 (2014).
8. Jia Y, et al. Gut microbiome modulates Drosophila aggression through octopamine signaling. *Nat. Commun.* **12**, 2698 (2021).
9. Gnainsky Y, et al. Systemic Regulation of Host Energy and Oogenesis by Microbiome-Derived Mitochondrial Coenzymes. *Cell Rep.* **34**, 108583 (2021).
10. Ben-Yosef M, Aharon Y, Jurkevitch E, Yuval B. Give us the tools and we will do the job: symbiotic bacteria affect olive fly fitness in a diet-dependent fashion. *Proc. Biol. Sci.* **277**, 1545-1552 (2010).
11. Yao Z, et al. Compartmentalized PGRP expression along the dipteran *Bactrocera dorsalis* gut forms a zone of protection for symbiotic bacteria. *Cell reports* **41**, 111523 (2022).
12. Gu J, Yao Z, Lemaitre B, Cai Z, Zhang H, Li X. Intestinal commensal bacteria promote *Bactrocera dorsalis* larval development through the vitamin B6 synthesis pathway. *Microbiome* **12**, 227 (2024).
13. Cai Z, et al. Intestinal probiotics restore the ecological fitness decline of *Bactrocera dorsalis* by irradiation. *Evolutionary applications* **11**, 1946-1963 (2018).
14. Raza MF, et al. Gut microbiota promotes host resistance to low-temperature stress by stimulating its arginine and proline metabolism pathway in adult *Bactrocera dorsalis*. *PLoS Pathog.* **16**, e1008441 (2020).
15. Sun X, Cui L, Li Z. Diversity and phylogeny of *Wolbachia* infecting *Bactrocera dorsalis* (Diptera: Tephritidae) populations from China. *Environ. Entomol.* **36**, 1283-1289 (2007).
16. Braig HR, Zhou W, Dobson SL, O'Neill SL. Cloning and characterization of a gene encoding the major surface protein of the bacterial endosymbiont *Wolbachia pipientis*. *J. Bacteriol.* **180**, 2373-2378 (1998).
17. Dedeine F, Boulétreau M, Vavre F. *Wolbachia* requirement for oogenesis: occurrence within the genus *Asobara* (Hymenoptera, Braconidae) and evidence for intraspecific variation in *A. tabida*. *Heredity (Edinb.)* **95**, 394-400 (2005).

18. Hoffmann AA, Clancy D, Duncan J. Naturally-occurring Wolbachia infection in *Drosophila simulans* that does not cause cytoplasmic incompatibility. *Heredity (Edinb.)* **76 (Pt 1)**, 1-8 (1996).
19. Roy S, Saha TT, Zou Z, Raikhel AS. Regulatory Pathways Controlling Female Insect Reproduction. In: *Annual Review of Entomology, Vol 63* (ed Berenbaum MR) (2018).
20. Toprak U, Hegedus D, Dogan C, Guney G. A journey into the world of insect lipid metabolism. *Arch. Insect Biochem. Physiol.* **104**, (2020).
21. Zhang J, et al. Identification of COP9 Signalosome Subunit Genes in *Bactrocera dorsalis* and Functional Analysis of *csn3* in Female Fecundity. *Front. Physiol.* **10**, 162 (2019).
22. Guo S, et al. Steroid hormone ecdysone deficiency stimulates preparation for photoperiodic reproductive diapause. *PLoS Genet.* **17**, e1009352 (2021).
23. Gao Q, et al. Noncanonical action of circadian clock genes controls winter diapause entry via the NuA4/TIP60 complex in *Harmonia axyridis*. *Proc. Natl. Acad. Sci. U. S. A.* **122**, e2510550122 (2025).
24. Cantó C, Menzies KJ, Auwerx J. NAD⁺ Metabolism and the Control of Energy Homeostasis: A Balancing Act between Mitochondria and the Nucleus. *Cell Metab.* **22**, 31-53 (2015).
25. Nunnari J, Suomalainen A. Mitochondria: in sickness and in health. *Cell* **148**, 1145-1159 (2012).
26. Twombly V, Blackman RK, Jin H, Graff JM, Padgett RW, Gelbart WM. The TGF-beta signaling pathway is essential for *Drosophila* oogenesis. *Development* **122**, 1555-1565 (1996).
27. Wahlström G, Norokorpi HL, Heino TI. *Drosophila* alpha-actinin in ovarian follicle cells is regulated by EGFR and Dpp signalling and required for cytoskeletal remodelling. *Mech. Dev.* **123**, 801-818 (2006).
28. Wang H, Jin L, Zhang H. Comparison of the diversity of the bacterial communities in the intestinal tract of adult *Bactrocera dorsalis* from three different populations. *J. Appl. Microbiol.* **110**, 1390-1401 (2011).
29. Tian Z, et al. Effect of host shift on the gut microbes of *Bactrocera cucurbitae* (Coquillett) (Diptera: Tephritidae). *Front. Microbiol.* **Volume 14 - 2023**, (2023).
30. Yang Y, Liu X, Xu H, Liu Y, Lu Z. Effects of Host Plant and Insect Generation on Shaping of the Gut Microbiota in the Rice Leafhopper, *Cnaphalocrocis medinalis*. *Front. Microbiol.* **13**, 824224 (2022).
31. Zhang B, Yang W, He Q, Chen H, Che B, Bai X. Analysis of differential effects of host plants on the gut microbes of *Rhopitoceros cyatheae*. *Front. Microbiol.* **15**, 1392586 (2024).
32. Blum JE, Fischer CN, Miles J, Handelsman J. Frequent replenishment sustains the beneficial microbiome of *Drosophila melanogaster*. *mBio* **4**, e00860-00813 (2013).
33. Shan H-W, et al. The plant-sucking insect selects assembly of the gut microbiota from environment to enhance host reproduction. *npj Biofilms and Microbiomes* **10**, 64 (2024).
34. Hershko A, Ciechanover A. The ubiquitin system. *Annu. Rev. Biochem.* **67**, 425-479 (1998).
35. Komander D, Rape M. The ubiquitin code. *Annu. Rev. Biochem.* **81**, 203-229 (2012).

36. Chen EH, Wei D, Wei DD, Yuan GR, Wang JJ. The effect of dietary restriction on longevity, fecundity, and antioxidant responses in the oriental fruit fly, *Bactrocera dorsalis* (Hendel) (Diptera: Tephritidae). *J. Insect Physiol.* **59**, 1008-1016 (2013).
37. Storelli G, et al. *Drosophila* Perpetuates Nutritional Mutualism by Promoting the Fitness of Its Intestinal Symbiont *Lactobacillus plantarum*. *Cell Metab.* **27**, 362-377.e368 (2018).
38. Jose PA, Ben-Yosef M, Jurkevitch E, Yuval B. Symbiotic bacteria affect oviposition behavior in the olive fruit fly *Bactrocera oleae*. *J. Insect Physiol.* **117**, 103917 (2019).
39. Quijano JC, et al. *lola* Is an Evolutionarily New Epigenetic Regulator of *dpp* Transcription during Dorsal-Ventral Axis Formation. *Mol. Biol. Evol.* **33**, 2621-2632 (2016).
40. Wang H, Jin L, Peng T, Zhang H, Chen Q, Hua Y. Identification of cultivable bacteria in the intestinal tract of *Bactrocera dorsalis* from three different populations and determination of their attractive potential. *Pest management science* **70**, 80-87 (2014).
41. Kaur M, Thakur M, Sagar V, Sharma R. Diversity of culturable gut bacteria of diamondback moth, *Plutella xylostella* (Linnaeus) (Lepidoptera: Yponomeutidae) collected from different geographical regions of India. *Mol. Biol. Rep.* **49**, 7475-7481 (2022).
42. Ranawat B, Mishra S, Singh A. *Enterobacter hormaechei* (MF957335) enhanced yield, disease and salinity tolerance in tomato. *Arch. Microbiol.* **203**, 2659-2667 (2021).
43. Tshishonga K, Serepa-Dlamini MH. Draft Genome Sequence of *Enterobacter hormaechei* Strain MHSD6, a Plant Endophyte Isolated from Medicinal Plant *Pellaea calomelanos*. *Microbiology resource announcements* **8**, (2019).
44. Ullah A, Mustaq H, Sh F, Manghwar H, Shah A, Chaudhary H. Plant growth promoting potential of bacterial endophytes in novel association with *Olea ferruginea* and *Withania coagulans*. (2017).

Reviewers' comments:

Reviewer #1 The authors have addressed my concerns and made the necessary changes. Therefore, I consider the modified paper suitable for publication.

However, I have one observation regarding the statistical significance in the tables of the Source Data file. The authors should be attentive when interpreting the pairwise comparisons. I identified three issues, but I recommend a thorough check before publication:

Answer: We thank the reviewer for carefully cross-checking the pairwise comparisons with the Source Data file. We regret any confusion caused by these inconsistencies. We have thoroughly checked all interpretations to ensure full consistency with the results reported in the Source Data file. We appreciate the reviewer's time and effort devoted to reviewing our manuscript.

1. Figure 1J: The comparison between Ctrl and ABX+EH shows no significant difference.

Answer: We thank the reviewer for the careful observation. We identified that an incorrect table for pairwise comparison was uploaded in the Source Data file for Figure 1J in the last version. However, the significance annotations in the figure and interpretations were correct reflecting the original experimental data. This issue has now been corrected in the "multiple comparisons test" section of the Source Data file based on the data, and we have ensured consistency among the figure, interpretations, and Source Data file. This does not alter any interpretations or conclusions of the manuscript.

2. Figure 1K (ovary): Only ABX and ABX+EH show a significant difference.

Answer: We thank the reviewer for pointing this out. As requested in the first round of review, we changed the statistical analysis to a nonparametric Kruskal–Wallis test based on the data distribution. Unfortunately, we missed to update Figure 1K (ovary) in the previous version of the manuscript after correcting statistical analysis. This has now been updated to be consistent with the Source Data file.

3. Figure 5L: There is a significant difference between ABX+dsLola1 and ABX+dsdpp.

Answer: We agree with the reviewer's observation. The Source Data file indicates a statistically significant difference between ABX+dsLola1 and ABX+dsdpp.

We have corrected the Figure 5L to include this comparison. This correction does not alter the interpretation of the results or the conclusions of the manuscript.

Additionally, in Figure 5I, the statistical analysis is missing.

Answer: We thank the reviewer for pointing this out. We have updated the Source Data file to include the corresponding statistical analysis result in the Source Data file.

As requested, we performed a thorough recheck. We corrected an inconsistency in the significance annotations in Fig. 4i and Supplementary Fig. 10b to ensure consistency with the corresponding Source Data file. We further verified that all figures, figure legends, and Source Data files are fully consistent. These revisions do not affect any interpretations of the results or the conclusions in manuscript. We are extremely grateful to the reviewer for highlighting the importance of careful checks.

Reviewer #2

I appreciate the authors' thorough and sincere response, including the additional data. The new analyses convincingly demonstrate that the gut microbiome is indeed extremely simple and stable. I also find the revised description of the "partial rescue" appropriate and mentioned. In addition, the recombinant protein injection experiments are straightforward and very well executed (Although I still have a minor concern regarding potential immune responses, this does not affect my overall assessment). Overall, these additional experiments and clarifications have fully addressed my previous concerns.

Answer: We thank the reviewer for the positive evaluation. We agree that recombinant protein injection could, in principle, trigger immune response. In our experimental design, injections were performed using the same protocol and included a control group injected with an equivalent amount GST-His, thereby controlling for other potential effects. As noted by the reviewer, this consideration does not affect the conclusion of the study. Once again, we appreciate the reviewer's effort in improving our manuscript.